

# Three dimensional soil organic matter distribution, accessibility and microbial respiration in macro-aggregates using osmium staining and synchrotron X-ray CT

Barry G Rawlins[1], Joanna Wragg[1], Christina Rheinhard[2], Robert C Atwood[2], Alasdair Houston[3], R. Murray Lark[1], and Sebastian Rudolph[1]

[1]British Geological Survey, Keyworth, Nottingham, NG12 5GG, UK
[2]Diamond Light Source, Harwell Science & Innovation Campus, Chilton, OX11 0DE
[3]SIMBIOS, Abertay University, 40 Bell street, Dundee DD1 1HG, UK

*Correspondence to:* Barry G Rawlins (bgr@bgs.ac.uk)

**Abstract.** The spatial distribution and accessibility of organic matter (OM) to soil microbes in aggregates – determined by the fine-scale, 3-D distribution of organic matter, pores and mineral phases – may be an important control on the magnitude of soil heterotrophic respiration (SHR). Attempts to model SHR at fine scales requires data on the transition probabilities between adjacent pore space and soil OM, a measure of microbial accessibility to the latter. We used a combination of osmium staining and synchrotron X-ray CT to determine the 3-D (voxel) distribution of these three phases (scale 6.6 µm) throughout nine aggregates taken from a single soil core (range of organic carbon (OC) concentrations 4.2-7.7 %). Prior to the synchrotron analyses we had measured the magnitude of SHR for each aggregate over 24 hours under controlled conditions (moisture content and temperature). We test the hypothesis that larger magnitudes of SHR will be observed in aggregates with shorter length scales of OM variation (i.e. more frequent, and possibly more finely disseminated, OM and a larger number of aerobic microsites).

After scaling to their OC concentrations, there was a six-fold variation in the magnitude of SHR for the nine aggregates. The distribution of pore volumes, pore shape and volume normalised surface area were similar for each of the nine aggregates. The overall transition probabilities between OM and pore voxels were between 0.02 and 0.03, significantly smaller than those used in previous simulation studies. We computed the length scales over which OM, pore and mineral phases vary within each aggregate using indicator variograms. The median range of models fitted to variograms of OM varied between 178 and 487 µm. The linear correlation between these median length scales of OM variation and the magnitudes of SHR for each aggregate was −0.42, providing some evidence to support our hypothesis. We require a larger number of observations to make a statistical inference. There was no evidence to suggest a statistical relationship between OM:pore transition probabilities and the magnitudes of aggregate SHR. The solid-phase volume proportions (45-63 %) of OM we report for our aggregates were surprisingly large by comparison to those assumed in previous modelling approaches. We suggest this requires further assessment using accurate measurements of OM bulk density in a range of soil types.



# 1 Introduction

In soil heterotrophic respiration (SHR) microbes utilise the carbon in soil organic matter (SOM) as an energy source, releasing gaseous $CO_2$ which accumulates in the soil at significantly larger concentrations than in the atmosphere (Hirano et al., 2003). Ultimately this excess $CO_2$ is released to the global atmosphere. It is important we understand the processes that determine

variations in the magnitude of SHR because it influences the flux of carbon dioxide ($CO_2$) from soils to the atmosphere, an important part of the global carbon cycle with major implications for global climate change (Cox et al., 2000). The turnover of SOM is also an important control on the cycling of other macronutrients, notably nitrogen.

It has been suggested that it is essential to understand the influence of microscale intra-aggregate heterogeneity of soil properties to ensure that organic matter (OM) mineralisation can be modelled effectively (Falconer et al., 2015). The majority

of soil microbial communities reside in pore networks within soil aggregates which are three-dimensional (3-D) agglomerations of mineral particles, varying in size, that form a hierarchy (Tisdall and Oades, 1982) with small, micro-aggregates (<250 μm) forming larger, macro-aggregates (> 250 μm). Soil aggregates consist of complex mixtures of SOM, mineral particles, pore space, microbes and moisture. The accessibility of SOM to microbial communities (substrate availability) is determined by the distribution of pores (Chenu et al., 2001; Negassa et al., 2015) which also also determines water potential and the flux

of oxygen. Soil matric potentials vary over short scales due to the varying size of pores, with microbes concentrated at the interfaces between air and water. Decomposition rates of SOM may therefore be influenced by moisture content (Moyano et al., 2012)), pore size and location within an aggregate (Killham et al., 1993), and also by temperature and substrate quality (Davidson and Janssens, 2006), and microbial properties (Li et al., 2015).

The majority of SOM utilised by soil microbes, the former both as large individual particles and more finely disseminated

material associated with minerals, occurs both on the surfaces of, and within, soil aggregates (Leue et al., 2010). In controlled laboratory experiments, the magnitude of SHR has been shown to vary considerably between soil aggregates (Kravchenko et al., 2015). It has been suggested that the location of OM within soil aggregates may be a significant factor governing the magnitude of OM mineralisation in soil (Dungait et al., 2012). Microsites for aerobic SHR occur where SOM and pores are adjacent to one another in soil aggregates, but to date their frequency and spatial distribution have not been established within

macro-aggregates. If a large proportion of intra-aggregate SOM is occluded by minerals so that microbes cannot utilise it, there will be fewer interfaces between SOM and pores and the magnitude of SHR for such aggregates may be smaller than those in which SOM is more accessible (a larger proportion of pore:SOM interfaces). In a recent study, Juarez et al. (2013) asserted that soil structure may be of limited importance in determining rates of SHR at the scale of the soil core. The authors created soils with differing structural properties (undisturbed versus disaggregated and sieved) and showed that after the structural

peturbations had dissipated, there were no significant differences in SHR rates for both native and added soil carbon. However, the observed increase in rates of SHR following disturbance also implies that soil structure does exert an influence on rates of microbial SOM minerlisation.

Another feature of soil aggregates that may influence the magnitude of SHR is the size and distribution of its SOM including particulate organic matter (Kravchenko et al., 2015). Consider two aggregates, with the same concentration of SOM, removed



from a single soil core. In the first aggregate the SOM consists of small, finely disseminated material that occurs frequently over short length scales, whilst in the second there are fewer, larger particles of SOM with larger distances separating them. We hypothesise that in the former aggregate there will likely be a larger number of microsites leading to a greater magnitude of SHR compared to the latter. This hypothesis could be tested by determining the magnitude of SHR in such aggregates if it

were also possible to subsequently determine the length scales over which the SOM is distributed in these aggregates. The 3-D spatial variation of SOM, mineral and pore phases in aggregates can be investigated using: i) geostatistics and ii) determining whether there are Representative Elementary Volumes (REVs) where properties are computed for increasing scales and in which smaller volumes are nested within larger volumes.

To date, relatively few experimental approaches have been applied to determine: i) the accessibility of soil OM in macro-

aggregates, and ii) whether the accessibility and of SOM in soil aggregates exerts a strong influence on microbial SHR at the macro-aggregate scale. This is in part because scientists have lacked methods for fine ($<10$ μm) scale 3-D discrimination between minerals and SOM within aggregates. Approaches to date have generally been limited to mapping SOM in two dimensions (Lehmann et al., 2007), or in 3-D within smaller regions of aggregates (Yu et al., 2016). In a recent study, Hapca et al. (2015) used a combination of X-ray computed tomography (CT) and scanning electron microscope images to map soil

chemical composition, including soil carbon, at a resolution of 15.8 μm in a small block of soil (side length of 1 cm). An alternative approach was recently demonstrated by Peth et al. (2014) where differences in X-ray absorption above and below the osmium (Os) edge — using synchrotron beamline X-ray computed tomography (CT) – was used to discriminate between OM, and mineral phases in aggregates of around 2-3 mm diameter, at a resolution of 9.77 μm.

In this paper we report the results of applying and extending the Os-staining and synchrotron X-ray CT method devel-

oped by Peth et al. (2014). Specifically we establish the 3-D distribution of mineral, SOM and pore space throughout nine macro-aggregates from a single soil core at fine (6.6 μm) length scales. To our knowledge such data have never been analysed geostatistically to determine the 3-D length scales over which SOM, minerals and pores vary both within and between aggregates. In so doing we establish the magnitude of any structural differences between the nine aggregates. Prior to the synchrotron X-ray CT analyses we measured the magnitude of SHR of each aggregate in the laboratory by measuring headspace $CO_2$ con-

centrations after incubating the aggregates in separate vials, having controlled for both temperature and moisture content. We have determined the accessibility of SOM within each aggregate by computing the transition probabilities of adjacent SOM and pore voxels, and also transition probabilities between the other phase combinations using the 3-D voxel classification. We used the SOM-pore transition probabilities as an index of SOM accessibility to determine whether it is strongly related to the magnitude of SHR for each aggregate, the latter scaled by total organic carbon (TOC) content. We also computed the length

scales over which SOM varies in each aggregate and compared this to their magnitudes of SHR, again scaled by aggregate TOC content. We discuss the implications of our findings for empirical and modelling studies which aim to (respectively) quantify and simulate the magnitude of SHR at the macro-aggregate scale.



## 2 Materials and methods

### 2.1 Aggregate samples, preparation and respiration measurements

An intact, cylindrical core of soil (diameter and length 50 mm) was collected from a field that had been under pasture for more than 15 years (British National Grid reference metres; Easting 463374, Northing 331992) in Keyworth (near Nottingham, UK).

The upper edge of the soil cylinder was inserted (at a depth of 5cm below the top of the mineral soil horizon) into a vertical soil face that had been exposed with a spade (the lower level of the cylinder was at a depth of 10cm below the top of the mineral horizon). The parent material of the soil at this site is a mudstone and the soil is a Luvisol (WRB, 2007) in texture-class Clay (Hodgson, 1974) based on a particle size analysis of material from a soil core collected from a location adjacent to the sampling site. After return to the laboratory (at room temperature) the intact core was removed from its container and placed on a plastic

sheet. The core was broken apart by hand to separate large aggregates along natural fracture surfaces and also those formed by fine roots. This procedure yielded aggregates of different sizes. We selected a subset of 9 aggregates which had shapes that were approximately either cubes or spheres with side length or diameter (respectively) of approximately 5-6 mm. We visually inspected each aggregate and rejected any which appeared to be dominated by a large single particle (e.g. a stone fragment).

We weighed each aggregate in a pre-weighed and labelled weighing boat and then placed each on a saturated 1 bar pressure

plate (Soil Moisture Equipment Corp, Santa Barbara, CA) for 20 minutes so that the moisture content of each aggregate would increase markedly. We then set the pressure plate to 0.5 bar ($-50$ kPa) for 5 hours so that each aggregate had the same moisture content, slightly less than field capacity. After removal from the pressure plate the aggregates were placed inside a filter insert (Costar (R) Spin-X Centrifuge Tube, Sigma-Aldrich, UK) which had been partially filled with 500 μm quartz beads (Sigma Aldrich, UK). A small quantity of quartz beads were placed on top of the aggregate to ensure each was surrounded by beads and

the insert (including quartz beads and aggregate) was re-weighed. The filter inserts were then placed inside a glass headspace vial (Thames Restek, Pennsylvania, USA) each of which had been filled with an equal quantity of quartz beads to reduce the volume of air in the vial. A crimping tool was used to seal the headspace vials with an aluminium seal and the vials were placed in an incubator at 37° C for 24 hours. After removal of the vials from the incubator, the concentration of $CO_2$ in the headspace of each vial was determined by removing a subsample of gas using a syringe and injecting it into a gas chromatograph (porapak

column Q 80/100 mesh in Agilent GC 7820) which had been calibrated with $CO_2$ standards of 100, 500, 1000 and 2000 mg $l^{-1}$. We subtracted the background concentration of $CO_2$ (400 mg $l^{-1}$) from each headspace analysis to give the excess $CO_2$ due to respiration. The septa and seals were then removed from the headspace vials and the aggregate and filter holder placed into a Costar (R) Spin-X Centrifuge Tube (total volume 2 ml; see Figure 1). The un-capped centrifuge tubes were placed in a freeze drying unit for 48 hours until any moisture in the vials had been removed completely. By removing all moisture from

the aggregates we wanted to ensure that the $OsO_4$ would diffuse completely through the soil aggregate pore space. The filter inserts were then reweighed so that the dry mass of each aggregate could be calculated.



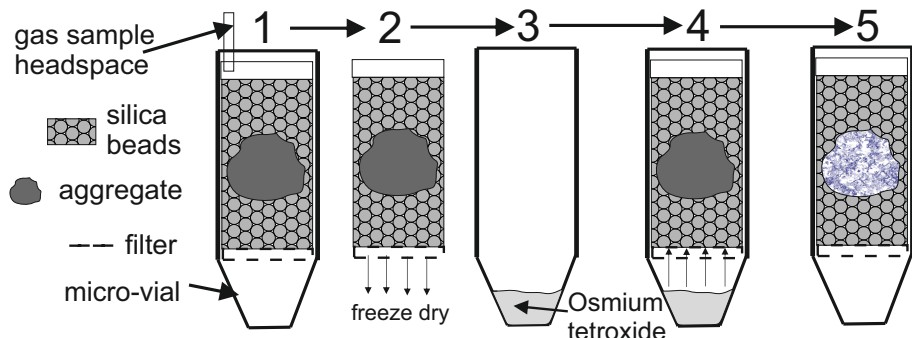

**Figure 1.** Schematic diagram showing a series of five steps in the treatment of the aggregates in their vials: 1) headspace gas removal, 2) freeze-drying, 3) & 4) staining with OsO$_4$ and 5) scanning in the synchrotron beamline.

## 2.2 Osmium staining of soil organic matter

The aggregates (each inside a filter insert and centrifuge vial) were placed inside a fume cupboard so that osmium tetroxide (OsO$_4$) could be used to stain the OM in each of the soil aggregates (Peth et al., 2014). A set of strict health and safety procedures were adopted due to the hazardous nature of OsO$_4$ but we do not describe these in detail here. Half a millilitre of

OsO$_4$ was pipetted into the bottom of each centrifuge vial and the vials sealed with caps. Each was left for 48 hours inside the fume cupboard during which time the Os diffused through the base of the filter, through the glass beads and Os was adsorbed preferentially by the carbon bonds in the OM. A schematic diagram showing the main steps in this procedure is shown in Figure 1.

The filter inserts were then removed and wiped clean and the top of each filter insert was sealed using caps and Araldite

resin. Each filter insert was then fixed to bespoke stainless steel supports using Araldite resin so that the filters could be placed into a synchrotron beamline.

## 2.3 Synchrotron X-ray CT analysis

Each of the nine aggregates inside the filter inserts were scanned using synchrotron X-ray CT at the Diamond Light Source (Harwell, UK) using the I12 beamline. Each aggregate was scanned at three energy levels: 53, 73.2 and 74.4 keV, the latter

were determined to be just below and above the K-absorption edge for Os, based on initial scanning of an osmium standard material. The 53 keV energy level provided an effective means of separating the solid and pore phases. The images for each horizontal slice through the aggregates were reconstructed yielding a set of 32-bit .tif files in which each pixel has an adsorption value for each energy level, and each pixel represents a 3-D voxel with side length of 3.3 µm.

## 2.4 Total Organic Carbon content of aggregates

After the aggregates had been scanned in the beamline they were carefully removed from their vials and their TOC content estimated using a Elementar Vario Max C/N analyzer at 1050 °C. Prior to measurement any inorganic carbon was removed




from the aggregates by adding HCl (5.7 M), then dried at 100 °C for 1 hour. The limit of quantification for TOC for a typical 300 mg sample was 0.18%

## 2.5 Synchrotron X-ray CT data processing

All the synchrotron X-ray CT data was processed using the same protocol described here. The first procedure was to subtract the absorption values from the 73.2 keV energy level (below the Os absorption edge) from the images created from the 74.4 keV energy level (above the Os absorption edge; see Peth et al. (2014)). This was undertaken using a script written in the R (R Core Team, 2013) and we subsequently refer to the resulting data and files as 'diffedge' (difference at the absorption edge). An example of the differences in absorption values above (74.4 keV) and below (73.2 keV) the Os absorption edge for one soil aggregate image slice example is shown in Figure 1. Note that the absorption values are generally larger above the Os edge, below the 1:1 line in Figure 2.

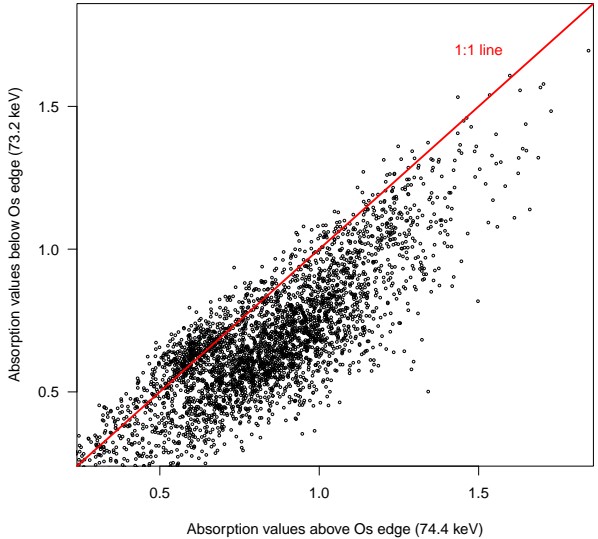

**Figure 2.** Raw absorption values above and below the Os absorption edge for one soil aggregate slice from the synchrotron X-ray CT beamline.

### 2.5.1 Creating masks for aggregate slices

Prior to analysing the synchrotron data at the three energy levels it was necessary to create masks of the aggregate outlines so that pixels outside the aggregate could be excluded from the analysis. Where quartz beads (with similar density to soil material) occurred adjacent to aggregates in the synchrotron images, those pixels within the beads were replaced with background absorption values using the software package VG Studio Max prior to creating masks for these images. To create the masks



we used a multi-stage procedure using scripts in the R environment and the image processing package Fiji (Schindelin et al., 2012):

– map the original 32-bit image into an 8-bit .bmp image and then apply a procedure for outlier removal written as an R script that was applied to all image slices of each aggregate.

– crop the image to extract a smaller sub-image that contains the aggregate using a macro written for Fiji.

– using a macro written in Fiji we segmented each image using statistical region merging (with parameter Q=between 10 and 25).

– apply binary segmentation to each image using the adjust threshold function to extract as much of the aggregate as possible.

– using the binary image we then used the 'fill holes' function in Fiji which provides a mask of the aggregate that contains

cracks.

– to fill in the cracks we wrote a majority filtering code in python and applied it several times until the cracks in the aggregate masks were completely filled.

Each mask was saved as an 8-bit .tif file. We wrote an R script using the raster package (Hijmans, 2014) to crop all the original images (nine aggregates each with two sets of images: i) 53 keV, and ii) the diff edge images) using the masks for each

associated image slice, setting all the values outside the cropped region to a constant value.

### 2.5.2   Reduction of raster resolution

The original synchrotron image files comprised 2544 × 2544 pixels with each pixel measuring 3.3 µm and and between 1400 and 2600 slices in each aggregate image stack. We found that it was challenging to process these data using large 3-D numeric arrays so we chose to reduce the resolution of the image stack by 50% in each dimension. By doing so we achieved an 8-fold

reduction (i.e. a 2-fold reduction in each dimension) in the size of the numeric array. Aggregation was carried out from top to bottom for each of two neighbouring layers (e.g. layer 1&2, 3&4, . . . ). The aggregation process was carried out in three steps. All these steps were applied to the images from the 56 keV energy level and the diffedge files. First the masks were applied to each file and adjacent layers were aggregated horizontally (2-D) by a factor of 2 using the aggregate function in the raster package (Hijmans, 2014) using the mean function to compute the mean of the layers. Then the two horizontally aggregated

layers were averaged vertically yielding a matrices of dimension 1272 × 1272. Any averaged outlying values greater than 10 were set to a value of 10. The matrices were combined into a 3-D numeric array with the third (vertical dimension) equal to the half the number of slices in the original data. These processing steps were undertaken using an MPI computer cluster.

### 2.5.3   Segmentation of solid and pore phases

We undertook exploratory analysis of the data in each horizontal slice of the 3-D numeric array at the 56 keV energy level. We

plotted the frequency distribution of the absorption values and observed two distinct but overlapping distributions, one with a





smaller mean value (pores) than the other (solid phase). To segment the solid and pore phase distributons in each aggregate slice we wrote an R script using the mixtools package (Benaglia et al., 2009), specifically utilising the *normalmixEM2comp* function which is a fast algorithm for two-component mixtures of univariate normal distributions. The algorithm requires starting values of the model parameters which, in this case, are the mean and variance of the data values for the pore and solid phase, and

their relative proportions. We assigned each pixel to either pore or solid phase based on the class with the largest posterior probability density under the mixture model. Using these classes we created a series of .tif image files storing the two-fold solid:pore classification, with a third class value for pixels outside the aggregate.

### 2.5.4   Porosity, surface area, pores sizes and shapes

We used the images from the segmentation of solid and pore phases to compute a series of physical properties for each ag-

gregate. Total porosity and surface area on planar/volume images were calculated using a bespoke Win32 computer program *minkowski.exe* that includes estimation algorithms published in Ohser and Mucklich (2000). The bespoke software was originally developed and verified by Alasdair Houston at SIMBIOS (Abertay University) as part of a research degree programme. We used the *3D Objects counter* plugin (Bolte and Cordelières, 2006) for Fiji to compute the volume of each pore in the maximum regular block of voxels that could be extracted from each of the nine aggregate arrays. We also used the surface area

and pore volume outputs from *3D Objects counter* to compute the pore shape factor (F) as:

$$F = \frac{Ae}{A} \tag{1}$$

where $A_e$ is the surface area of a sphere with a volume equal to that of the pore and A is the measured pore surface area (Wadell, 1932). A value of 1 for the F parameter represents a sphere, whilst smaller F-values refer to more irregular or elongated pore shapes.

### 2.5.5   Separation of mineral and organic matter phases

Prior to the measurement of TOC, we had removed all the water from the aggregates by freeze-drying so we can compute the mass of mineral matter ($M_m$) by subtracting the mass of OM (two × TOC content; (Pribyl, 2010)) from the total mass of the aggregate. We also know the bulk density of mineral matter ($BD_m$) in soil to be 2.65 g cm[3] (Hall et al., 1977) so we calculated the volume of mineral material ($V_m$) in each aggregate as:

$$V_m = \frac{M_m}{BD_m} \tag{2}$$

We were then able to compute the volume of OM by subtracting the volume of mineral material from the total volume of the solid phase (mineral & organic matter; section 2.5.3) that was computed using the Gaussian mixture model. In this way we were able to determine the proportions of OM and mineral volumes in the solid phase. Given that we know the volume of OM ($V_{om}$) in each aggregate and its mass ($M_{om}$; total aggregate mass minus the mineral mass), we can compute the bulk density

of the OM ($BD_{om}$) in each aggregate as:

$$BD_{om} = \frac{M_{om}}{V_{om}} \tag{3}$$





In doing so we were able to check that the bulk density values for OM were consistent with previously published data, taking into account any differences in moisture content. To achieve our objective of separating all the solid phase voxels into either mineral or OM classes we used: i) the diffedge data values for each aggregate, and ii) the proportions of mineral and OM volumes in each aggregate. Those voxels with the largest values in the diffedge numeric array for each aggregate were assigned

an OM classification, and the proportion of solid phase voxels assigned as OM was the volume proportion of OM in each aggregate (see Table 1). The other solid voxels were assigned a mineral classification.

## 2.6 Statistical and geostatistical analyses

### 2.6.1 Transition probabilities

We assume that the critical locations in soil for microbial respiration are the interfaces between (voxels of) OM and pore space

(Monga et al., 2008). We ask the question: how many such voxel interfaces are there per unit volume of soil? We define an OM/pore interface voxel as an OM class voxel from which one can make a one-step transition to a pore voxel. For the present we ignore direction of transition. The (transition) probability that a voxel is at an OM/pore interface is the probability that a pair of adjacent voxels are OM and pore, which we refer to as $P(OM|pore)$. We computed transitions probabilities for an arrangement where we consider the 26 voxels around a central OM voxel. Consider a cube with side length of 3 voxels, giving

a total of $3^3 = 27$ voxels in total, with $27 - 1 = 26$ transitions from the central voxel. We use the transition probability between OM and pore as a quantitative measure of OM accessibility.

We computed transition probabilities for the 3-D numeric arrays in which each voxel was one of four classes: 0=mineral ($M$), 1=pore ($pore$), 2=organic matter ($OM$) and 9=mask. The total of the three transition probabilities ($P(pore|pore)$, $P(M|pore)$ and $P(OM|pore)$) is one. We wrote a script in R that progressed from the from top to the bottom of the numeric array, starting

from the second layer and ending at the penultimate layer. For every iteration three neighbouring layers were used (one above, a central layer, and one below) avoiding the outermost rows and columns of the 3-D array in the analysis. All three layers were simultaneously shifted by one pixel around their initial position in the $x$ and $y$ directions, while for every offset combination the voxel classification was queried and concatenated with the classification of the original non-shifted central layer which shared the same spatial location ($x$ and $y$). The class comparisons always originated from the central voxel to either the six main facing

voxels or the complete $3 \times 3 \times 3$ array subset. For the six main face voxels we computed transition probabilities for all nine phase combinations, including transitions to and from the same phase. For the 26 neighbouring voxels, we only considered transitions from central OM voxels. In addition to the voxel class, we also recorded the direction of transition was because: i) this was necessary to remove the combination where the central layer was compared with the non-shifted central layer, and ii) in future analyses (not in this paper) we may wish to undertake directional analyses of transitions. We computed the frequency

of each class combination (transition) for each layer comparison and in a final step the frequency of each transition and layer comparison was computed.

In a recent study, Falconer et al. (2015) used the proportion of OM-centred voxels which had at least one transition to a pore voxel in adjacent voxels (in our case there were 26) as a measure of OM accessibility and so we computed these proportions for



our nine aggregates. To distinguish this additional measure of accessibility from the transition probabilities, we refer to these former values as the minimum threshold OM-pore proportions.

### 2.6.2 Indicator variograms and variogram models

To understand the length scales over which the three phases vary, we computed 3-D indicator variograms (Webster and Oliver, 2007) for each phase and for each aggregate using scripts written in R with the *gstat* package (Pebesma, 2004). Using the 3-D numeric arrays in which each voxel had been classified as either mineral, organic matter, pore or mask (exterior) we chose a random starting point within the numeric array of each aggregate and selected a cube of voxels with side length 200 around this point. We then checked that less than 10% of the voxels were classed as mask. If mask values accounted for more than 10% of all voxels, a new random starting point was selected until this condition was met. We converted this 3-D array into a *gstat* object ($x$, $y$ and $z$ coordinates plus the voxel class, and excluded the exterior voxels). To compute indicator variograms for each phase we recoded the phase classes so that a single phase took a value of one, and the other phases were set to zero. We then randomly selected a subset of 50 000 voxels with which to estimate the indicator semi-variances for each phase at a series of increasing lag intervals up to a maximum of 250 voxels. We plotted a set of indicator variograms for each phase and fit a range of single authorised variogram models to them (Webster and Oliver, 2007) . In all cases the exponential model gave the best fit and so we computed and recorded the parameters of the exponential model fitted to each set of indicator semi-variances. For the model range parameter, we recorded the effective range which in the *gstat* package is three times the theoretical range reported. We repeated this procedure 50 times for each of the three phases in each of the nine aggregates to ensure that we encompassed the variation inside each aggregate. We note that each indicator variogram may not be independent because in some cases the starting points may be sufficiently close for the 3-D arrays of side length 200 to overlap.

## 3 Results and their interpretation

### 3.1 Aggregate properties and respiration rates

A set of properties determined for each of the nine soil aggregates are summarised in Table 1. There was an approximate two-fold variation in the quantity of TOC in the nine aggregates (range 4.18-7.7%), whilst the magnitude of respiration (based on the excess $CO_2$ concentration scaled to the TOC content) varies by more than a factor of 6 (range 10.5-65.9 µg C mg $C^{-1}$). There was an approximate three-fold variation in total porosity between the aggregates (range 4.4-11.1%) but no clear relationship between total porosity and the magnitude of respiration scaled by TOC content. At the scale of the soil core we have observed strong statistical relationships between topsoil bulk density and the square root of TOC for this soil type in a local cultivated field (Lark et al., 2014), but there was no similar relationship for our nine aggregates taken from a single core. In addition, there was no strong statistical relationship between total porosity and bulk density. The bulk density values for the nine aggregates (range 1.29-1.83 g $cm^{-3}$) were generally larger than would be predicted from a pedotransfer function that uses the TOC content of soil cores (Lark et al., 2014), but this can be explained by the markedly larger porosity values (20-45%) that





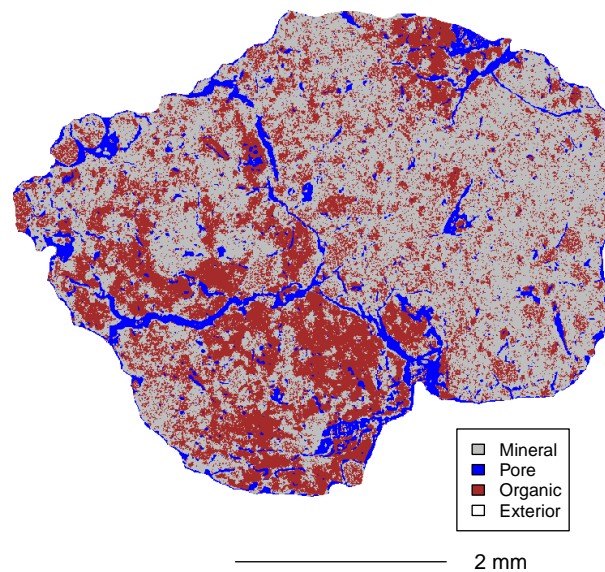

**Figure 3.** An example horizontal cross-section through one of the aggregate (number 40) showing the distribution of mineral, pore and organic matter phases. Movies showing sequential images of each aggregate slice will be posted as supplementary material to the manuscript.

we reported at the soil core scale (Lark et al., 2012) from a cultivated field on the same soil type. There was limited variation in the surface area of the aggregates after it has been normalised by aggregate volume (range 10.1-15.6 mm$^2$ mm$^{-3}$; see Table 1).

Figure 3 shows distribution of the OM, pore and mineral phases in one aggregate slice. Using our approach to computing the volume of mineral and organic matter in each of the aggregates (section 2.5.5) it is noteworthy that the organic matter comprises a large proportion of the solid volume (48.5-63.4%). Also, the dry organic matter bulk density values (following freeze drying) are larger (range 0.35-0.81 g cm$^{-3}$) than those estimated by Adams (1973) from soil cores (0.207-0.311 g cm$^{-3}$) over moisture contents ranging from field capacity ($-10$ kPa) to a water potential of $-20$ kPa. The value of 0.81 g cm$^{-3}$ for aggregate number 76 is anomalous and we cannot account for it based on the other physical properties we report in Table 1.

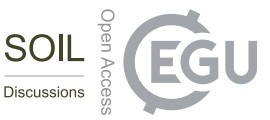

**Table 1** Physical properties of the nine soil aggregates and quantities of $CO_2$ released by the aggregates from soil heterotrophic respiration during incubation (see text).

| Aggregate number | 37 | 40 | 43 | 49 | 55 | 61 | 67 | 73 | 76 |
|---|---|---|---|---|---|---|---|---|---|
| Dry mass (mg) | 106 | 118 | 177 | 103 | 112 | 177 | 146 | 101 | 103 |
| Aggregate volume (mm$^3$) | 73.7 | 70.2 | 137.3 | 70.8 | 79.2 | 118.3 | 95.8 | 60.4 | 56.4 |
| TOC (%) | 6.81 | 5.27 | 7.06 | 5.53 | 7.7 | 6.63 | 6.83 | 4.18 | 7.48 |
| Dry bulk density (g cm$^{-3}$) | 1.43 | 1.68 | 1.29 | 1.46 | 1.41 | 1.50 | 1.52 | 1.67 | 1.83 |
| [a] Porosity (%) | 11.1 | 7.72 | 7.29 | 8.88 | 7.01 | 4.42 | 5.64 | 6.86 | 7.31 |
| [b] Mineral mass (mg) | 91.2 | 105.5 | 152.2 | 91.7 | 94.8 | 153.8 | 125.9 | 92.6 | 87.7 |
| [c] Dry OM bulk density (g cm$^{-3}$) | 0.44 | 0.50 | 0.35 | 0.38 | 0.45 | 0.42 | 0.47 | 0.40 | 0.81 |
| [d] OM volume (%) | 51.1 | 61.4 | 44.6 | 53.8 | 48.5 | 51.1 | 52.6 | 62.1 | 63.4 |
| [d] Mineral volume (%) | 48.9 | 38.6 | 55.4 | 46.2 | 51.5 | 48.9 | 47.4 | 37.9 | 36.6 |
| [e] Moisture loss (freeze-drying) (%) | 23.5 | 18.1 | 27.8 | 21.3 | 25.3 | 25.8 | 25.2 | 22.8 | 20.0 |
| [f] Surface area (mm$^2$) | 1042 | 706 | 1397 | 895 | 1011 | 1336 | 1242 | 891 | 881 |
| Surface area/agg. volume (mm$^2$ mm$^{-3}$) | 14.1 | 10.1 | 10.2 | 12.6 | 12.8 | 11.3 | 13.0 | 14.7 | 15.6 |
| [g] Excess headspace C concentration (mg kg$^{-1}$) | 76 | 108 | 238 | 350 | 223 | 477 | 656 | 127 | 301 |
| [h] Normalised C gas concentration (µg C mg C$^{-1}$) | 10.5 | 17.3 | 19.1 | 61.5 | 25.8 | 40.6 | 65.9 | 30.1 | 39.0 |

[a] solid:pore thresholds computed using two-component Gaussian mixture model from adsorption values at 56keV (see text).

[b] Mineral mass computed from total aggregate mass − mass of organic matter ($2\times$ TOC).

[c] Organic matter (OM) bulk density (g cm$^{-3}$) computed using Equation 3.

[d] Expressed as a proportion of the solid volume (excludes pore space) and assumes a mineral density of 2.65 g cm$^{-3}$.

[e] The mass proportion (%) of moisture lost between mass of field-moist aggregates and freeze-dried mass (the latter after changes in moisture introduced through the pressure plate). We cannot compute volumetric moisture content (cm$^3$ cm$^{-3}$) because the freeze-drying procedure also removes moisture from both pore space and organic matter.

[f] Surface area was computed using a bespoke programme (see text).

[g] A background $CO_2$ concentration of 400 mg kg$^{-1}$ was assumed.

[h] Headspace C gas concentration normalised by the TOC content of each aggregate.



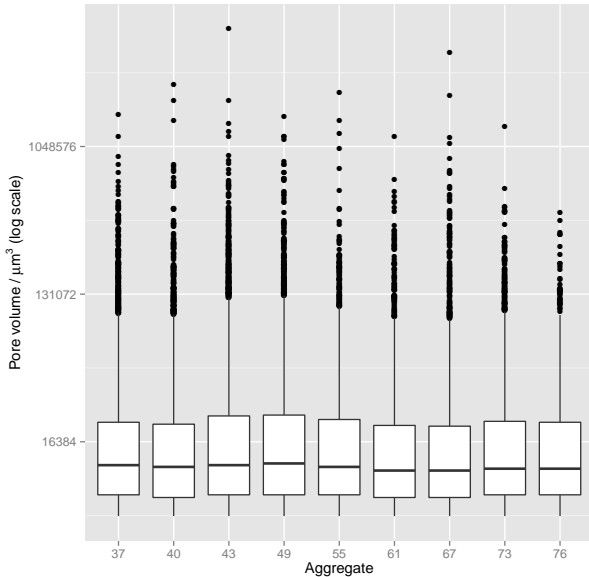

**Figure 4.** Distribution of pore volumes ($\mu m^3$) in a regular 3-D region extracted from each of the nine aggregates.

## 3.2 Porosity, surface area, pore sizes and shapes

Figure 4 shows the variation in pore volumes ($\mu m^3$) computed for the regular regions extracted from each of the nine aggregates in the form of a boxplot. For the smaller pores ($<30\,000\ \mu m^3$) the distributions are very similar (median value range 10924-12074 $\mu m^3$), whilst there are a greater number of large pores ($>100\,000\ \mu m^3$) in some aggregates compared to others. The pore

shape factor (F) distribution is also similar across all the aggregates (Figure 5), with some aggregates having a greater number of the most elongated pores ($F < 0.2$) than the others. Overall we infer that both pore size and elongation are relatively similar for each aggregate and therefore are unlikely to account for significant differences in the magnitudes of SHR we observed.

## 3.3 Transition probabilities

Figure 6 shows the variation in the overall transition probabilities from a central voxel to the adjacent 26 faces for the nine

aggregates. As expected, the largest overall transitions are between the same phases with overall probabilities of between 0.64 and 0.94. The smallest transition probabilities are from mineral and organic phases to the pore phase (range 0.01-0.05) which reflects the smaller total proportion of pore voxels compared to the two other phases. Although on average the proportion of organic voxels is larger than the mineral voxels (see Table 1), the median transition probability from pore to mineral (0.17) is larger than the median transition probability from pore to organic (0.11).

Figure 7 shows the overall transition probabilities from each organic matter central voxel to the neighbouring 26 voxels for each of the nine aggregates (in a restricted region of a ternary diagram). The largest overall transition probability in each case





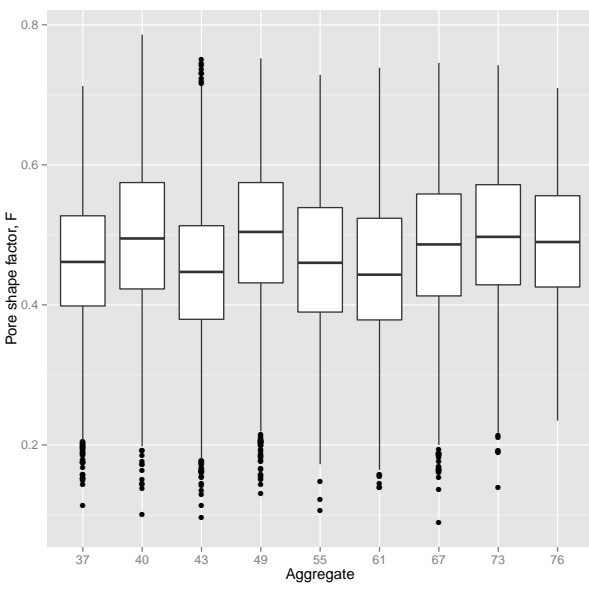

**Figure 5.** Variation in the pore shape factor (F) in a regular region extracted from each of the nine aggregates .

is from:to the organic phase (range 0.66-0.75) whilst the overall organic to mineral phase transitions were between 0.22 and 0.31. In terms of OM accessibility, the OM-pore transition probabilities are small transition (range 0.02-0.03) which indicated that only a small proportion of all OM voxels are accessible to soil microbes. The values of our alternative measure of OM accessibility was the proportion of OM-centred voxels with at least one adjacent pore voxel (the minimum threshold OM-pore

5   proportions) are shown in Table 2. These values range from 13.1 to 19.0% showing that the frequency distribution of the numbers of adjacent pore voxels is positively skewed, with a larger frequency of voxels having only one OM-pore transitions compared to larger numbers of OM-pore transitions around a central OM voxel (overall transition probability range 2.2-3.1 %). There was no significant relationship between the magnitude of SHR (rescaled to aggregate TOC content) and the transition probabilities from a central OM voxel to a neighbouring pore voxel (Figure 8).

**Table 2** - The proportion of minimum threshold voxels (at least one OM-centred voxel has an interface with an adjacent pore voxel) in each soil aggregate.

| Aggregate | 37 | 40 | 43 | 49 | 55 | 61 | 67 | 73 | 76 |
|---|---|---|---|---|---|---|---|---|---|
| Proportion (%) | 16.0 | 17.2 | 13.1 | 16.4 | 13.8 | 15.1 | 15.0 | 19.0 | 14.3 |




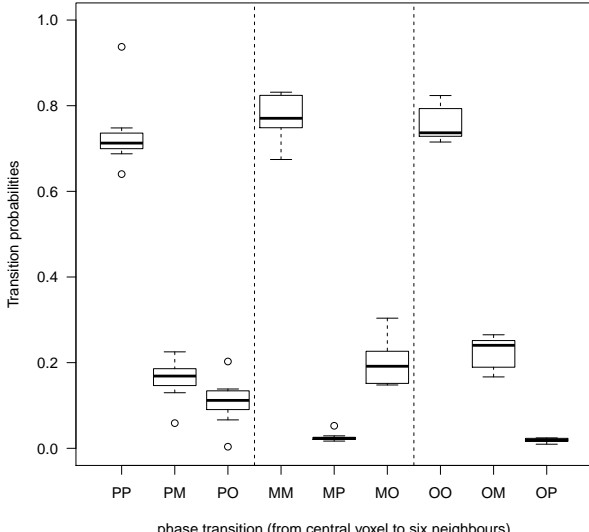

**Figure 6.** Boxplot showing the transition probabilities between a central voxel of organic matter and the 26 neighbouring voxels for each of nine (numbered) aggregates. The three transition types are: i) OO=organic matter to organic matter, ii) OP=organic matter to pore and iii) OM=organic matter to mineral. Note how the scales are truncated to a restricted region of the ternary diagram.

### 3.4 Indicator variograms and models fitted to them

Figure 9 shows the variation of the effective range of the exponential models fitted to the indicator variogram semivariance estimates for the 50 randomly selected blocks in each of the aggregates. Selected statistics of the effective range values are shown in Table 3. In seven of the aggregates the median effective range of porosity is less than the mineral and organic matter phase effective ranges, whilst in two of the aggregates (40 & 49) the pore phase has a larger effective range. In three of the aggregates (49, 55 & 61) the interquartile range (IQR) of the effective range values are generally smaller than the other aggregates. The IQR of the effective range of the mineral and organic matter phases are generally larger than the pore phase, and the median effective range values are also generally more variable than the pore phase (Figure 9). Given the considerable differences in the length scales of OM variation between the nine aggregates, we considered there may also be equivalent differences in the frequency of microsites for microbial respiration, with sites occurring more frequently where OM varies over shorter length scales. In Figure 10 we show the median of the effective model ranges (Table 3) of OM matter plotted against the TOC normalised respiration values from the laboratory measurements (Table 1). Although there are only nine pairs of observations, there is evidence of a negative relationship between respiration rate and length scale of OM variation (Pearson linear correlation $=-0.42$) . We consider this evidence to support our original hypothesis concerning microsites and length scale of OM variation, but we need a larger set of observations to make a statistical inference.



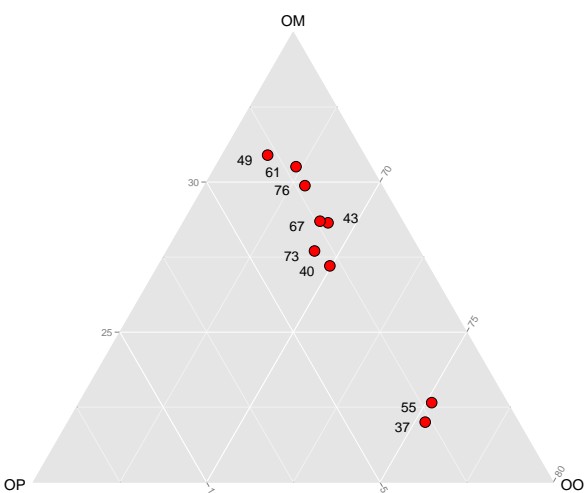

**Figure 7.** Ternary diagram showing the overall transition probabilities (%) between a central voxel (side length 6.6 μm) and 26 adjacent voxels (see text) for each of the nine aggregates (labelled). The nine transitions are between each of the three phases (O=organic matter; P=pore, M=mineral). For example, OO=organic matter to organic matter. Note the truncated axes show only a sub-region of the ternary diagram.

We could not find other published studies which have reported the relative magnitude of unaccounted for short-scale (6.6 μm) variation of mineral and OM phases in three dimensions within soil aggregates, and these values may be useful for similar studies. This unaccounted for variation (nugget variance) is the variance of analytical error plus variation that occurs at scales shorter than the sampling resolution. The magnitude of nugget variance is often expressed as a proportion of the nugget plus

5   sill variance (the variance at the range of the fitted model). Across all aggregates, the mean and median proportions of nugget variance for the mineral and organic phases was 0.56 (56%) whilst the mean and median proportions of the pore nugget variance was smaller, 0.23 (23%).

## 4   Discussion

We have reported what we understand is the first data showing complete 3-D macro-aggregate scale distributions of OM,

10   pore and mineral phases at fine (6.6 μm) scales, plus the length scales over which they vary and the transition probabilities at interfaces between these phases. These data could be used to test both existing and new models which aim to account for small (aggregate) scale variations in SHR (Monga et al., 2008; Falconer et al., 2015). To fully understand the processes governing



**Figure 8.** Scatterplot showing the transition probabilities between a central voxel of organic matter and 26 neighbouring voxels versus the magnitude of respiration (normalised to aggregate TOC content).

**Table 3** - Selected statistics of the effective exponential model ranges ($\mu$ m) fitted to the indicator variograms of organic matter voxels for each of the nine aggregates.

| Aggregate | 25th percentile | median | 75th percentile | sd |
|---|---|---|---|---|
| 37 | 316 | 458 | 647 | 229 |
| 40 | 217 | 289 | 2681 | 1607 |
| 43 | 222 | 301 | 407 | 519 |
| 49 | 182 | 193 | 207 | 18 |
| 55 | 358 | 388 | 419 | 51 |
| 61 | 159 | 178 | 195 | 30 |
| 67 | 255 | 330 | 465 | 700 |
| 73 | 410 | 487 | 581 | 211 |
| 76 | 370 | 449 | 503 | 1325 |

SHR in the soil aggregates we studied it is necessary to quantify the distribution of OM and pore space at scales of less than 250 μm, the scale below which the majority of variation in the these key soil properties occurs.

Our analyses could be used to improve quantitative estimates of the accessibility of both particulate ($> 50$ μm) and finer organic matter in soil aggregates. Our analysis showed that OM voxels which had an overall transition probability to a neighbouring pore voxel (our criteria of accessibility) of between 0.02 and 0.03. In their simulation study, Falconer et al. (2015) used



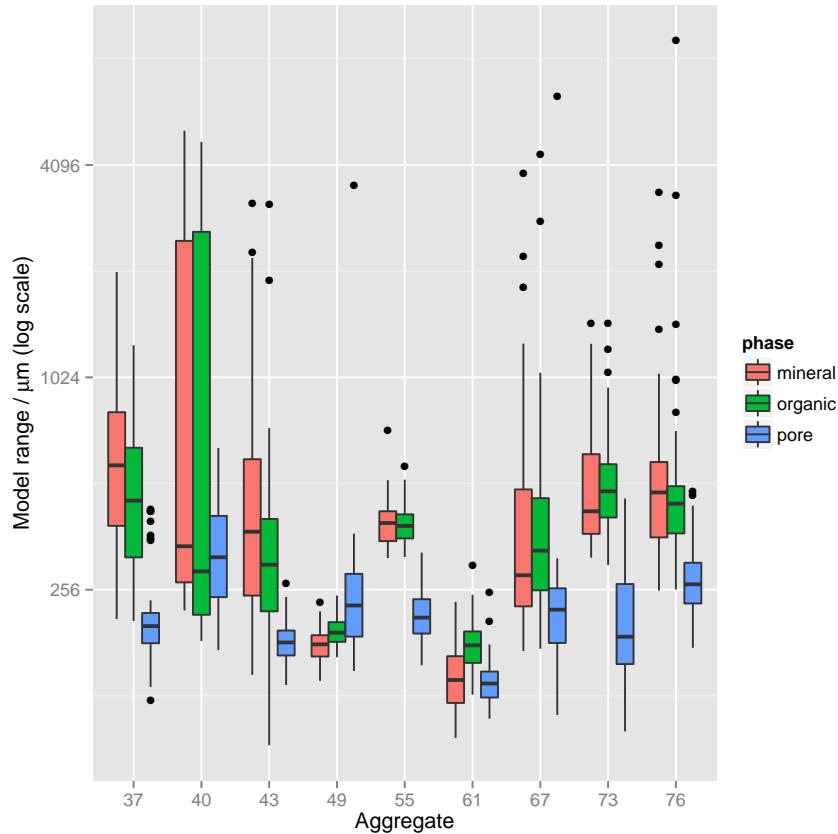

**Figure 9.** Boxplot showing variations in the model range estimate for exponential models fitted to indicator variograms of the three phases (mineral, soil organic matter and pore space) for each of the nine aggregates. The semivariance estimates were computed from a subsample of 50 000 locations from 50 separate blocks (each measuring $200 \times 200 \times 200$ voxels) within each aggregate.

proportions of accessible particulate OM voxels (those with at least one neighbouring pore voxel) of between 20 and 100 % (range of OM contents 1.4-7 %). Using the same metric, we estimated that accessible OM voxels account for a substantially smaller proportion (range 13.1- 19 %) of the total quantity of OM in each of the nine aggregates. In their simulation study Falconer et al. (2015) placed POM at the pore:solid interface which likely accounts for the larger particulate OM accessibility proportions they report and this approach may require modification if more realistic POM accessibility proportions are to be simulated.

We plan to extend our analysis further using the approach proposed by Kravchenko et al. (2015) by: i) quantifying the distribution of particulate organic matter ($> 50$ μm) and, ii) identifying those pores which are connected to the exterior of each aggregate, providing a direct pathway for the diffusion of gas to and from sites of intra-aggregate SHR. These data could be used to assess whether these two factors are strongly correlated with the magnitude of SHR. It would also be helpful to



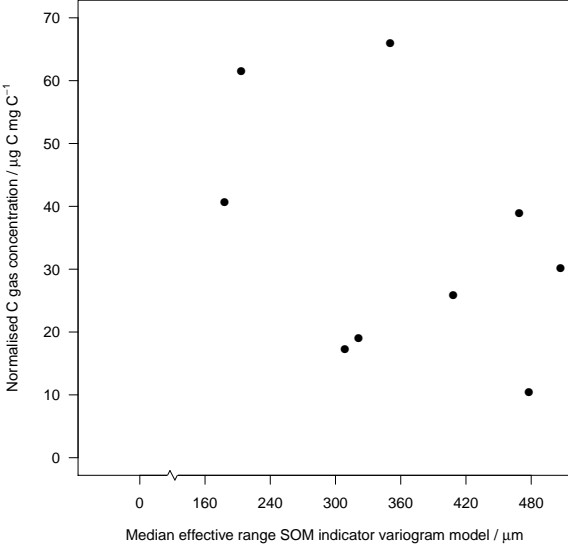

**Figure 10.** Scatterplot of median model range estimate for exponential models fitted to indicator variograms of organic matter versus headspace $CO_2$ gas concentration normalised to the carbon content of the aggregate.

estimate the distribution of water-filled pores (Monga et al., 2008) at the suction (−50 kPa) used in our experiment as this would determine which sites of microbial respiration may have been anaerobic, influencing the magnitude of SHR.

We note that the OM volumes (%) computed for our aggregates were large (range 44.6-63.4%) given their TOC content (range 4.2-7.5 %). In a recent modelling approach (Falconer et al., 2015), the authors estimated various physical properties of

soil aggregates including OM volumes of between < 1 % and 3.35 % for TOC contents of between 0.7 and 3.5 % (assuming 50 % of OM is organic carbon based on Pribyl (2010)). If we compare these TOC contents and OM volumes reported by Falconer et al. (2015) with those we report, the former are significantly smaller. After a comprehensive search we could only find values for OM bulk density reported by Adams (1973) from soil cores and we suggest this property requires further investigation across scales to ensure that realistic values are used to estimate soil bulk density and OM volumes in studies of

soil aggregates.

We reported that for a set of nine aggregates there was some evidence that shorter length scales of OM variation were associated with larger magnitudes of SHR, having scaled the respiration rates to TOC content. This relationship requires further investigation using macro-aggregates with a wide range of TOC concentrations and textures to determine whether this relationship is statistically significant, and whether it supports our interpretation relating to the more frequent occurrence

of microbial microsites. We might expect aggregates from more finely textured soil, in which greater quantities of OM are preserved (Angers et al., 2011), to have shorter scales of OM variation than those soils of coarser texture.





In their study, Peth et al. (2014) used air-dried soil aggregates whilst we chose to freeze-dry the aggregates in our experiment to maximise absorption of osmium onto organic carbon bonds. It would be useful to compare the results of both approaches to determine whether the drying process has significant implications for the extent of Os absorption throughout an aggregate. This could be achieved by comparing the magnitude and spatial distribution of Os using energy dispersive X-ray analysis linked to

a scanning electron microscope, the method used by Peth et al. (2014) to validate their original approach.

## 5   Conclusions

We have shown how a combination of synchrotron X-ray CT, osmium staining and TOC measurements can be used to successfully quantify the 3-D distribution of OM, pore and mineral phases throughout soil macro-aggregates at fine scales (6.6 μm). The magnitude of SHR which we measured for each of nine macro-aggregates (controlling for moisture content and tempera-

ture) varied by a factor of six whilst their TOC contents varied by less than a factor of two. Many of the physical properties of the nine aggregates were very similar: the pore size and shape distributions, and surface area normalised by aggregate volume.

The transition probabilities between OM-centred voxels and adjacent pore voxels – a measure of OM accessibility – were both small and of limited variation (probabilities between 0.02 and 0.03) for all aggregates and there was no clear relationship between accessibility and the magnitude of SHR. There were substantial differences in median length scales (median ranges

178-487 μm) over which OM varied between aggregates based on models fitted to their indicator variograms. We contend that shorter length scales of OM variation leads to greater frequencies of microsites and greater $CO_2$ production through microbial respiration and we present preliminary evidence to support this relationship. Further research is required to investigate the strength of this relationship in aggregates with a wider range of soil OM contents and for differing soil textures. We believe these are the first data to quantify 3-D macro-aggregate OM accessibility at fine scales and could be used to help parameterise

models of OM mineralization.

*Author contributions.*   The following authors (initials) made the specified contributions to the paper: BGR coordinated the project, lab work, data analysis and wrote parts of the manuscript; CR and RA undertook the synchrotron analyses of the osmium-stained soil aggregates and contributed to the synchrotron analyses in the manuscript; RML contributed to the design of the experiment, the geostatistical analysis of the data and wrote part of the manuscript; AH provided the code for computing surface area and contributed to the manuscript. JW undertook

the laboratory-based osmium staining and headspace $CO_2$ analyses and wrote the methods section; SR wrote R codes for the analyses of transition probabilities of the 3D arrays and other analyses and wrote this section of the manuscript.

*Acknowledgements.*   This work was supported by the use of the Diamond Beamline (I12) at the Diamond Synchrotron (Harwell, UK) project reference number EE9535. We thank Martin Hurst for helping to create the aggregate masks and Craig Sturrock for assistance with the image processing. We thank the following staff from BGS for their assistance with the beamtime at Diamond: Toni Milodowski, Jeremy Rushton,

Lorraine Field and Dan Parkes. We thank Jemma Purser for undertaking the headspace $CO_2$ analyses and Vicky Moss-Hayes for measuring





the TOC content of the nine aggregates. Simona Hapca (Abertay University) devised the approach to create masks for each aggregate slice.

This paper is published with the permission of the Executive Director of the British Geological Survey (NERC).



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
