# Peer review of "Three dimensional soil organic matter distribution, accessibility and microbial respiration in macro-aggregates using osmium staining and synchrotron X-ray CT"

_SOIL, 2016_

## Referee Comment (RC1) · Anonymous Referee #1 · 13 May 2016

This is a very interesting and innovative study on a novel and exciting topic of relationships between soil organic matter and pores in soil macro-aggregates. The authors are using a novel osmium-staining CT-scanning based technique to identify SOM and are utilizing very sophisticated tools for in-depth spatial analyses of the image data. The manuscript certainly warrants publication, however, there is a need for some revisions.

My main concern that must be addressed before the manuscript is published is the method of pore size characterization used in the study. It appears that the researchers used an object identification algorithm to identify individual pores and then used volumes and a shape factor of the identified pores as one of the main tools in characterizing them. I am afraid I have to say that this approach is quite meaningless, and probably some of the lack of pore effects reported in the study is just reflecting the fact that inadequate criteria of pore characterization were used. This approach completely ignores pore diameters and tortuosity - that is, the pore characteristics that are most relevant to their functioning. Say, we identified two pores with approximately the same volumes. One of them could be a thin and long tortuous pore, the other can be a large round cavity. Their functioning in terms of water, air, microbes, SOM decomposition, anything, will be completely different, yet in the classification system of this study they will be lumped in the same size class. While the distance from the pore component of the study is valid and interesting, the components that are based on the object-based pore identification (Figs. 4 and 5) should not be included in the manuscript.

Minor items: p.3 l. 6 -the part regarding representative volume does not seem to fit with the rest of the study.

p.3 l. 10 - something is missing after "and"

The experimental part seems to be very thoroughly conducted. I am curious - what was the need in using glass beads? Not having them would simplify the authors life a lot in terms of creating aggregate masks.

Are all these details in describing how the aggregate masks were created really needed? A lot of the steps talk about in-house R codes or macros and, without those provided as part of the manuscript, this procedure description is not something that anybody from the audience can even try to reproduce.

I think it is unfortunate that the authors decided to aggregate the image data. Why not just use the subsections of the original 3 micron resolution data sets?

While I do not see it as a big problem for the current study I believe in future the authors should seriously consider the need to look not just at pores in general, but to keep in

mind that depending on their diameters and other characteristics pores can function very differently. The authors expectations regarding pore-emission-SOM relationships that are expressed in the manuscript are not reasonable for many pore types/sizes.

I agree that scaling CO2 emission by TOC makes sense, but just for "quality" check - was there a positive correlation between SHS and TOC? Because if everything worked as expected there should be one, and it would be nice to hear about it. If there was none, it is also important to report.

p.10 l.30 The discussion on differences between aggregate and bulk soil findings is a bit simplistic. It is a basic soil science knowledge that density of aggregates is typically greater than soil bulk density (simply put, soil bulk consists of aggregates and large pores among them). Much more interesting would be comparisons of porosity, density, etc. results of this study with literature data that were collected on the same spatial scale (i.e., based on aggregates).

Please take a look at the following source for assessments of soil organic matter density:

Mayer, L.M., L.L. Schick, K.R. Hardy, R. Wagai, and J. McCarthy. 2004. Organic

matter in small mesopores in sediments and soils. Geochim. Cosmochim.

Acta 68:3863–3872. doi:10.1016/j.gca.2004.03.019

In Table 1 and in other places that mention porosity it should be noted that here we are looking at image-based porosity that reflects volume of pores above certain threshold.

I am very excited about lower OP probabilities results of this study. To me it is an indirect indication of pore presence to be conducive to OM decomposition.

Fig.6 - maybe do not show PP, OO, and MM values? They are not informative and without them the differences in other transition groups will be more visible.

I have to admit that what is shown on Fig.7 and its relationship to what is shown on

Fig.6 eludes me.

Figs. 8 and 10 - even though the relationships are not significant, adding regression line, p-value and r2 would be good.

I understand the driving for reporting the probabilities as the main outcome of this study from the modeling perspective, but can this probability information be somehow presented in units of actual distances? I believe it would be of interest to greater audience.

---

## Referee Comment (RC2) · Anonymous Referee #2 · 3 Jun 2016

The article deals with 3D visualization and quantification of organic matter (SOM) in soil aggregates. It is an interesting topic to many SOM researchers. The authors stained SOM in aggregates with Osmium (Os) tetroxide and scanned the aggregate using synchrotron X-ray CT. In general the article is casually written with an elaborate method and out of focus discussion section. Abstract should focus on the key message in concise form. The last paragraph of the introduction section should be brief with clear objectives. Discussion section should include validity of the experimental approach, justification of results obtained (i.e. porosity, pore shape, SOM volume, accessibility and soil respiration).

[Figure]

The authors followed largely the staining and scanning protocol published by Peth et al. (2014). The authors haven't provided any experimental data to demonstrate that Os was preferentially taken up by SOM only not adsorbed on mineral matrix of the soil. The authors used aggregates from a Clay soil for their experiment. Low diffusivity of clay soil could preclude the flow of Os vapour to SOM but increase the chance of adsorption of the vapour on clay surfaces. Moreover, Os can also react with clay-SOM complex not only the particulate SOM (POM) in the aggregates. From the Figure 3 it is not at all clear (resolution is too coarse) whether Os adsorbed on mineral matrix or SOM or POM. In my view, much better presentation could be a thresholded image slice showing pores and SOM alongside with greyscale scanned image of that slice. It will be nice to see if the authors could separate the 3D distribution of SOM adsorbed on clay surfaces and POM. Another concern, POM and adsorbed SOM both contain carbohydrates, will this affect Os reaction with SOM? I think the methodological approach followed in this work requires a calibration/verification protocol. Authors could use X-ray spectroscopy to verify the SOM distribution they found in an image slice using Os staining and scanning. A standard sample with known distribution of SOM or POM can also be used to verify the method presented in this paper. Authors presented that SOM occupied >50% of total aggregate volume, although %SOM was 4-7%, which is very difficult to grasp and warrant a validation of the approach used. Authors also need to present concentration of POM and SOM on silt+clay particles in their aggregates to justify the 3D distribution of SOM.

Authors also need to present a thresholded image and greyscale scanned image to demonstrate their stepwise approach of image segmentation. Authors need to describe how the pores and stained SOM separated during phase segmentation of the image slices. Since the volume of SOM was calculated by subtracting volume of mineral phase from total volume of soil solid phase, accuracy of the image thresholding is very important. Authors also referred 2.65 g cm-3 as bulk density of the mineral matter but should be written as particle density of the mineral particles. Moreover, the term density of organic matter is much preferable than "bulk density" of organic matter.

The figures presented in the article are not clear enough to show the distribution of pore geometry in the aggregates. The naming of 9 aggregates in Tables and Figures is not clear. A graph with multiple lines showing pore volume against pore diameter in different aggregates, I think would be much more informative than presenting Figure 4 as boxplots. Figure 5. Is it possible to extract images of different pore shapes of aggregates using threshold pore images? Authors can use threshold images to demonstrate the variation in pore shape and then distribution of different shapes in aggregates. Figure 6: Authors should focus on transition between SOM and pores. I feel it would much better if the authors could translate transition probability values in a form understandable for wider audience. Not clear why Figure 7 is included in the text. Figure 8 and 10: Dull scatter plots, a simple regression equation with R2 value can covey the massage. If possible calculate pore connectivity from the dataset and plot it against SHR. Table 3: not clear why this table is needed. Authors need to present variogram model graphs showing the spatial variability of SOM in the aggregates. The graphs are more informative than the presented box plots in Figure 9.

Authors incubated aggregate samples in 37°C for 24 hours and then measured the $CO_2$ concentration of the headspace. The temperature was bit high to measure soil respiration and I suppose it gradually made the aggregates dry over 24 hours, which would affect the respiration rate.

The authors wrote in many instances they used custom wrote scripts/macros in R and Fiji without presenting the codes. Authors may present the codes in supplementary material of the manuscript.

---

## Author Response (AR2)

**Comment 1:** *P4:25 provide volume of subsample taken for CO2 analysis*

**Response**: Done as suggested.

**Comment 2:** *P8:l6 value instead of values.*

**Response**: Done as suggested.

**Comment 3:** *P8:l5 We then computed the threshold adsorption value at 56 keV that equated to the transition from pore (smaller adsorption) to solid phase (larger adsorption) Im sorry but I dont really understand. Does this mean that you empirically determined the threshold grey value based on local minima in the summed histogram of the entire aggregate? Or is the threshold theoretical by comparing X-ray attenuation of air and mineral matter? Or did you per aggregate set the threshold grey value so that the total number of voxels assigned as pores equated to Vp / volume of one voxel? Needs further explanation.*

**Response**: We have used your wording to make this clearer.

**Comment 4:** *P10:l13 excess CO2 concentration sounds awkward. Would be better to use just omit (based on the excess CO2 concentration scaled to the TOC content) from this sentence. The actual unit of your reported respiration is not g C mg C-1 but in fact g C mg-1 day-1. Consider revising..*

**Response**: Revised as suggested.

**Comment 5:** *p13:4-7 Difficult sentence to follow. Please rephrase (perhaps split in two).*

**Response**: We have changed this to make it clearer.

**Comment 6:** *Discussion/conclusion: At present only a 24h incubation has been used to estimate microbial utilization of the native SOM present in the aggregates. It is well known that cumulative C-mineralization usually follows a non-linear course. Firstly: SOC is inherently composite and exists of a continuum of decomposability and so after depletion of C respired at a high rate from labile constituents within a few days, only slower respiration indicated by a more gradual CO2 emission following a 0-order kinetic is usually observed. In the present study we are merely looking at a quantification*

*of the breakdown of the most labile parts. Their content may well have differed among the studied aggregates. Likely this is then what dominantly determined the variation in total respiration over 24h. It is then also evident that only weak correlations with quantifications of pore structure or physical distribution of the OM were found. Secondly, it is well known that drying leads to death of microorganisms and a peak C-mineralization flush after rewetting is related to the microbial processing of this necromass. Again the time frame of the soil incubations was likely too short to have a representative measure of microbial utilization of the majority of OM present. Alternative mechanisms explaining transient effects of soil moisture changes on microbial activity have been grouped under the term Birch effect. I would ask the authors to be more critical in their discussion and conclusion on the relatively limited approach used to assess microbial processing of native SOM.*

**Response**: We have added a sentence regarding the lability of SOM to the Discussion. By email we agreed that the Birch effect was not relevant because we did not dry the soils significantly prior to the SHR measurements.

---

## Author Response (AR3)

**Comment 1:** *My main concern that must be addressed before the manuscript is published is the method of pore size characterization used in the study. It appears that the researchers used an object identification algorithm to identify individual pores and then used volumes and a shape factor of the identified pores as one of the main tools in characterizing them. I am afraid I have to say that this approach is quite meaningless, and probably some of the lack of pore effects reported in the study is just reflecting the fact that inadequate criteria of pore characterization were used. This approach completely ignores pore diameters and tortuosity - that is, the pore characteristics that are most relevant to their functioning. Say, we identified two pores with approximately the same volumes. One of them could be a thin and long tortuous pore, the other can be a large round cavity. Their functioning in terms of water, air, microbes, SOM decomposition, anything, will be completely different, yet in the classification system of this study they will be lumped in the same size class. While the distance from the pore component of the study is valid and interesting, the components that are based on the object-based pore identification (Figs. 4 and 5) should not be included in the manuscript.*

**Response**: Our aim was to compare the overall size and shape characteristics of the nine aggregates and we did this with the two sets of analyses presented in the original manuscript. We agree that analyses of both pore diameter and tortuosity could be useful in terms of understanding pore functioning and we have computed these data using: i) the Fiji plugin *AnalyseSkeleton* for tortuosity index (computed as length of pores divided by Euclidean distance between their furthest ends) and ii) pore diameter using the *thickness* function in the *BoneJ* package. We have included an interpretation of these data in the manuscript. We disagree with the reviewer when he/she suggests that the data on pore size and shape should not be included (Figures 4 & 5). We consider these data are essential in the characterisation of the pores in the nine aggregates and we would wish to include them in the final version along with the new information. We have updated our interpretation to include the pore tortuosity and thickness data in relation to soil heterotrophic respiration.

**Comment 2:** *Minor items: p.3 l. 6 -the part regarding representative volume does*

*not seem to fit with the rest of the study.*

**Response**: Inclusion of this reference to representative elementary volumes was in error - it will be removed from the final version of the manuscript.

**Comment 3:** *p.3 l. 10 - something is missing after 'and'*

**Response**: The 'and' has been omitted from the final version and this now makes the sense clear.

**Comment 4:** *The experimental part seems to be very thoroughly conducted. I am curious - what was the need in using glass beads? Not having them would simplify the authors life a lot in terms of creating aggregate masks.*

**Response**: We included the quartz beads to prevent the aggregates from fragmenting. The greatest risk of this was at the freeze-drying stage, when the aggregates were subject to forces that could cause them to move and fragment inside the vials; and also during transport to and from the synchrotron. We could not select aggregates with diameters that were exactly the same as the vials (fixed diameter) and so any movement of the vial could cause collisions between the dry aggregates and the wall of the vial, causing them to fragment. The quartz beads acted as an inert supporting medium reducing the forces of fragmentation on the aggregates, ensuring their structure was maintained prior to synchrotron X-ray CT scanning. If we repeat this experiment, we may consider a more dense supporting material, such as stainless steel beads, to avoid the problems associated with making masks of the aggregates where surrounding material is of a similar density. However, it is possible that the larger density steel beads could also lead to fragmentation of the aggregates during transit. This needs to be tested further.

**Comment 5:** *Are all these details in describing how the aggregate masks were created really needed? A lot of the steps talk about in-house R codes or macros and, without those provided as part of the manuscript, this procedure description is not something that anybody from the audience can even try to reproduce*

**Response**: The stages and macros we refer to in this section could be reproduced

quite quickly in Fiji or R by any other researcher. If we did not provide these details they would not be able to do this and we considered it important that others could reproduce our workflow. We therefore chose to leave these detailed instructions in the final version of the manuscript and provided the scripts as supplementary material (see comment by reviewer 2).

**Comment 6:** *I think it is unfortunate that the authors decided to aggregate the image data. Why not just use the subsections of the original 3 micron resolution data sets?*

**Response**: We did not wish to aggregate the data. However, we chose to do so because of the difficulties in analysing the data in terms of computer memory and processing time. To undertake various analyses of the 3D numeric array at the original resolution ($2544 \times 2544 \times$ *ca.* 1200) was extremely challenging even using a high performance computing cluster. Preliminary tests showed that some analytical steps would take a number of days to complete and given this was to be repeated for 9 aggregates we took the decision to reduce the resolution so we could complete our study in a timely manner. Once reduced in resolution (by a factor of 8) each 3D numeric array was 340 Mb in size which we found to be manageable in terms of reading and writing from memory.

**Comment 7:** *While I do not see it as a big problem for the current study I believe in future the authors should seriously consider the need to look not just at pores in general, but to keep in mind that depending on their diameters and other characteristics pores can function very differently. The authors expectations regarding pore-emission-SOM relationship.*

**Response**: No response needed here we believe.

**Comment 8**: *I agree that scaling CO2 emission by TOC makes sense, but just for "quality" check - was there a positive correlation between SHR and TOC? Because if everything worked as expected there should be one, and it would be nice to hear about it. If there was none, it is also important to report.*

**Response**: Yes there was a positive correlation between SHR ($CO_2$ generated) and

TOC with a Pearson correlation of $r=0.29$. We have reported this in the new version of the manuscript but we do not place too great an emphasis upon it because we consider scaling $CO_2$ to TOC content to be a more relevant measure.

**Comment 9:** *The discussion on differences between aggregate and bulk soil findings is a bit simplistic. It is a basic soil science knowledge that density of aggregates is typically greater than soil bulk density (simply put, soil bulk consists of aggregates and large pores among them). Much more interesting would be comparisons of porosity, density, etc. results of this study with literature data that were collected on the same spatial scale (i.e., based on aggregates).*

**Response**: We do not agree that this discussion is too simplistic. We felt it necessary to explain these differences; not all readers will be familiar with the relationships between aggregate and larger scale bulk density values. The problem with comparing bulk densities (BD) of aggregates (with data captured at a similar scale) is that there are many features that can influence BD (SOC content, texture, mineralogy, differences in soil formation processes) and making clear interpretations concerning them would be problematic. We do not propose to change the final version of the manuscript in this respect.

**Comment 10:** *In Table 1 and in other places that mention porosity it should be noted that here we are looking at image-based porosity that reflects volume of pores above certain threshold.*

**Response**: We agree with this comment and will make these changes to the final version of the manuscript.

**Comment 11:** *Fig.6 - maybe do not show PP, OO, and MM values? They are not informative and without them the differences in other transition groups will be more visible.*

**Response**: We do not agree that the PP, OO and MM values should be removed from Figure 6. We consider it is helpful to show that the three phase transitions (for each phase) must sum to 1 and their inclusion makes this clear in the Figure. We have not

made this change to this Figure in the final manuscript.

**Comment 12:** *I have to admit that what is shown on Fig.7 and its relationship to what is shown on Fig.6 eludes me*

**Response**: Figure 7 shows the individual aggregate transition probability values of the far right panel (OO-OM-OP) of Figure 6 (here expressed as percentages rather than decimal proportions). This change in presentation style may have caused some confusion so we have altered Figure 7 to make this clear by reporting the transitions as decimal proportions in the final version. We have altered the captions to make the relations clearer.

**Comment 13:** *Figs. 8 and 10 - even though the relationships are not significant, adding regression line, p-value and r2 would be good*

**Response**: We refer the reviewer to an article on the use and misuse of regression by R. Webster (Webster, R. 1997. Regression and functional relations. *European Journal of Soil Science*, **48**, 557–566). Properly applied, regression is used to derive a predictive relationship of one variable from another or (in strictly limited circumstances) to calibrate a linear functional relationship. The regression line is not a suitable summary of the bivariate relationship between two variables unless one is measured without error. The regression line is therefore not a suitable decoration for the scatter plot. We can calculate the correlation coefficient, and report a $p$-value for the null hypothesis that is is zero, but this will reflect little more than the small size of our sample.

**Comment 14:** *I understand the driving for reporting the probabilities as the main outcome of this study from the modeling perspective, but can this probability information be somehow presented in units of actual distances? I believe it would be of interest to greater audience.*

**Response**: The transition probabilities are reported for one voxel transitions and the scale of the voxels is of side length of 6.6 µm. We only report and discuss these one-step transitions; transitions over larger scales could be computed and presented and we assume this is what the comment is suggesting, but we are not certain. We have

not undertaken these analyses to date and so we do not present them.

**Comment 1:** *Abstract should focus on the key message in concise form. The last paragraph of the introduction section should be brief with clear objectives. Discussion section should include validity of the experimental approach, justification of results obtained (i.e. porosity, pore shape, SOM volume, accessibility and soil respiration).*

**Response**: We consider that the original version of the abstract does focus on the key message, and does so concisely with clear objectives. In order to be self-explanatory the abstract requires some context for the study and not just results. The discussion section in our original version of the manuscript focussed on the wider implications of our findings, putting them in the context of other work and exploring options for further analyses of our data. We have included as part of our new discussion section consideration of how we might proceed to determine the location and quantities of finely disseminated organic matter sorbed onto mineral surfaces as suggested by reviewer 1 in other comments below.

**Comment 2:** *The authors followed largely the staining and scanning protocol published by Peth et al. (2014). The authors haven't provided any experimental data to demonstrate that Os was preferentially taken up by SOM only not adsorbed on mineral matrix of the soil. The authors used aggregates from a Clay soil for their experiment. Low diffusivity of clay soil could preclude the flow of Os vapour to SOM but increase the chance of adsorption of the vapour on clay surfaces. Moreover, Os can also react with clay-SOM complex not only the particulate SOM (POM) in the aggregates. From the Figure 3 it is not at all clear (resolution is too coarse) whether Os adsorbed on mineral matrix or SOM or POM. In my view, much better presentation could be a thresholded image slice showing pores and SOM alongside with greyscale scanned image of that slice. It will be nice to see if the authors could separate the 3D distribution of SOM adsorbed on clay surfaces and POM..*

**Response**: Peth et al. (2014) demonstrated using the same methodology that Os was preferentially adsorbed to organic matter rather than clay minerals. We do not consider it necessary to repeat the same verification steps as Peth et al. (2014). In addition, as we freeze-dried our samples we consider there is more scope for the Os

vapour to diffuse into finer pores than may have been the case with the Peth et al approach in which small quantities of moisture would have remained in the finest pores following their use of air-drying. The separation/identification of SOM adsorbed on clay surfaces is beyond the scope of this paper and we do not state it as one of our objectives. The osmium retention by clay was addressed in our original manuscript by basing the threshold for organic matter classification – using differences between Os absorption above and below the adsorption edge – on the inferred volumetric organic matter content of the aggregate. This requires that SOM adsorbs Os more than does clay, not that there is no adsorption of Os by clay. See further response to this in comment 5 below. In the final (modified) version of the manuscript we present a thresholded image for the same slice as shown in the original Figure 3, as suggested by the reviewer.

**Comment 3:** *Another concern, POM and adsorbed SOM both contain carbohydrates, will this affect Os reaction with SOM? I think the methodological approach followed in this work requires a calibration/verification protocol. Authors could use X-ray spectroscopy to verify the SOM distribution they found in an image slice using Os staining and scanning. A standard sample with known distribution of SOM or POM can also be used to verify the method presented in this paper.*

**Response**: In their paper, Peth et al. (2014) stated: 'We selected osmium as a staining agent as this reacts with unsaturated C-bonds of organic compounds ....including finely disseminated organic matter often absorbed onto clay mineral surfaces and not visible as discrete organic particles (Chenu and Plante, 2006). They showed that they could detect Os-staining of both POM and finely disseminated SOM and validated the method using SEM-based EDX X-ray analysis (see Figures 3 and 5 in Peth et al (2014)). Therefore we do not consider it necessary to undertake another validation of the approach.

**Comment 4:** *Authors presented that SOM occupied >50% of total aggregate volume, although %SOM was 4-7%, which is very difficult to grasp and warrant a validation of the approach used.*

**Response**: The approach we developed to estimate the volume of organic matter in each aggregate was based on sound physical principles and accurate estimates of constants (such as the density of mineral matter ($2.65$ g cm$^{-3}$)). The reviewer does not state on what basis he considers this approach to be flawed and without further detail we contend that our approach is justified and does not require further validation.

**Comment 5:** *Authors also need to present concentration of POM and SOM on silt+clay particles in their aggregates to justify the 3D distribution of SOM.*

**Response**: Some of this was also addressed in response to comment 2. We consider this to be beyond the scope of our paper. To our knowledge such an analysis has not been undertaken before and would require careful development in terms of the approach. We would need to identify an upper threshold size/volume/shape for sorbed organic matter and rules governing the size and shape of neighbouring mineral particles to select SOM on clay or silt particles. We have not modified our manuscript in this respect.

**Comment 6:** *Authors also need to present a thresholded image and greyscale scanned image to demonstrate their stepwise approach of image segmentation.*

**Response**: We have updated Figure 3 to show the thresholding, stepwise approach in the revised version of the manscript.

**Comment 7:** *Authors need to describe how the pores and stained SOM separated during phase segmentation of the image slices. Since the volume of SOM was calculated by subtracting volume of mineral phase from total volume of soil solid phase, accuracy of the image thresholding is very important.*

**Response**: We gave a detailed description of the segmentation of the image slices in the original version of the manuscript. We first separated the pores from the solid phase using a two component mixture algorithm. We then computed the the volume of organic matter and the differences in the adsorption values above and below the osmium adsorption edge to differentiate organic matter from mineral phases. The use of image thresholding based on the two-component mixture algorithm has a clear

theoretical basis for its application.

**Comment 8**: *Authors also referred 2.65 g cm⁻³ as bulk density of the mineral matter but should be written as particle density of the mineral particles. Moreover, the term density of organic matter is much preferable than 'bulk density' of organic matter.*

**Response**: We agree with this comment and we have amended the final version of the manuscript to reflect this.

**Comment 9:** *The figures presented in the article are not clear enough to show the distribution of pore geometry in the aggregates. The naming of 9 aggregates in Tables and Figures is not clear..*

**Response**: We believe this comment is directed at Figures 4 and 5. We needed a way to summarize the features of the pores (size and shape factor) and we consider that the boxplots presented do this effectively. We have undertaken further analyses of pore tortuosity and thickness/diameter and we present these new data in the final version of the manuscript (see response to comment 1 by reviewer 1).

**Comment 10:** *A graph with multiple lines showing pore volume against pore diameter in different aggregates, I think would be much more informative than presenting Figure 4 as boxplots.*

**Response**: This analysis is not possible based on the outputs from the 3D objects counter function from the BoneJ package which is routinely used for pore analysis. We consider that the boxplots in Figures 4 and 5 are informative as they summarize the data for each aggregate. We have also computed pore diameters for all 9 aggregates in response to comment 1 of reviewer 1 and we include these data in our modified version of the manuscript.

**Comment 11:** *Figure 5. Is it possible to extract images of different pore shapes of aggregates using threshold pore images? Authors can use threshold images to demonstrate the variation in pore shape and then distribution of different shapes in aggregates.*

**Response**: We described how we applied the 3D objects counter function in BoneJ (see section 2.5.4) to extract pore volume and surface area to compute pore shapes

for each pore structure from a regular block within each aggregate. This is based on the pore:solid phase threshold images. Our aim here was to summarise the overall features of the pore size and shape for each aggregate so they could be compared and we consider that this was achieved effectively.

**Comment 12:** *Figure 6: Authors should focus on transition between SOM and pores. I feel it would much better if the authors could translate transition probability values in a form understandable for wider audience.*

**Response**: We do focus on the transitions between organic matter and pores in Figure 7 (see next comment). We describe how to compute transition probabilities in our Methods section and we consider that the majority of readers would be able to understand this based on the mathematical notation which is not particularly complex.

**Comment 13:** *Not clear why Figure 7 is included in the text*

**Response**: Figure 7 presents, in a more detailed form than Figure 6, the transition probabilities between organic matter centred voxels (O) and the other phases. We consider this plot is useful as the reader can see clearly how these important properties, which have never been computed at the aggregate scale before, vary between the nine aggregates. Note we have improved this Figure based on a comment by reviewer 1.

**Comment 14:** *Figure 8 and 10: Dull scatter plots, a simple regression equation with R2 value can covey the massage.*

**Response**: See our response to Reviewer 1's comment 13. We are sorry that the reviewer finds the scatter plot dull, but the use of regression lines for decoration (except in some circumstances which do not apply here) is statistically unsound. We do include the correlation coefficient, however. We have not changed the final version of the manuscript in this respect.

**Comment 15:** *If possible calculate pore connectivity from the dataset and plot it against SHR.*

**Response**: We consider this to be beyond the scope of the current paper but could be addressed in a subsequent analysis.

**Comment 16:** *Table 3: not clear why this table is needed. Authors need to present variogram model graphs showing the spatial variability of SOM in the aggregates. The graphs are more informative than the presented box plots in Figure 9.*

**Response**: We disagree with the reviewer on this point. We chose to focus on the range parameter of the variogram models because this is the main feature of the spatial variation. We considered that an effective way to summarize the range data for each of the nine aggregates and three phases was by presenting a boxplot of the data and we consider that these present the data very effectively. Individual models for each region of each aggregate would confuse the reader in our view. We have not changed the final version of the manuscript.

**Comment 17:** *Authors incubated aggregate samples in 37C for 24 hours and then measured the CO2 concentration of the headspace. The temperature was bit high to measure soil respiration and I suppose it gradually made the aggregates dry over 24 hours, which would affect the respiration rate.*

**Response**: Based on the literature we considered 37 °C to be an appropriate temperature for incubation. As the vials were sealed during the incubation phase we do not expect the soils would have dried substantially over this period.

**Comment 18:** *The authors wrote in many instances they used custom wrote scripts/macros in R and Fiji without presenting the codes. Authors may present the codes in supplementary material of the manuscript.*

**Response**: Yes we can provide these as supplementary materials.

Biochemistry, 78, 189  194.

**To: Steven Sleutel**

**From: Barry Rawlins - lead author**

**Date: 27th Sept 2016**

Dear Steven

My co-authors and I have re-processed the data based on an assumed density of organic matter of 1.4 g cm$^{-3}$ which was one of the main criticisms of the original version. This approach is detailed in the new version of the manuscript. By doing so we created a completely new partition of the three phases (mineral, pore and organic matter) in each of the nine aggregates. There is evidence to suggest this was effective because we now observe a strong negative correlation ($r = -0.98$) between aggregate bulk density and porosity. I am confident this has significantly improved the paper which we now think is ready for further consideration. In revising the new manuscript we addressed all the other comments made in the two referee reports that we referred to in our original responses. I have also responded to each of your topical editor comments below:

**Comment:** *My major concern is the one as R2Comm4: The volume % of organic matter in the studied aggregates was unusually large. But much seems to be related to the spatial resolution of 3µm. To most readers it would appear that the organic matter fraction was strongly overestimated by the adopted Os-staining procedure and that OM densities of 0.4g cm-3 are extremely low. Usually adopted values are 1.4 (e.g. Chenu et al., 2006 EJSS). Is this discrepancy explained by the simple fact that you are considering bulk densities of OM? This would imply that much of the aggregate volume classified as OM in fact includes air inside sub-resolution pores as well. So the question remains how much of the aggregate volume was approximately overestimated to be either mineral or organic because pores <3 µm were misclassified as being OM or mineral matter? This needs to be addressed still in the manuscript and it should be*

*stipulated that the OM densities in table 1 represent apparent bulk densities.*

**Response**: We have re-processed the data using an OM density of 1.4 g cm$^{-3}$ and we think this has significantly improved the accuracy of the estimates for the three phases. We therefore no longer present OM density values in Table 1.

**Comment:** *I concur with the authors that no new proof needs to be presented for the validity of the Os-staining procedure (R2Com2 & 3). Nevertheless the authors should acknowledge that Os-staining of OM sorbed onto mineral particles could have resulted in misclassification of voxels that are partly OM and partly mineral matter as being entirely organic. There is no way of confirming this because such edge effects are only visible at finer spatial resolutions. This knowledge should, however, be taken along when critically assessing the quantified volumes of organic phase and OM densities. Most likely the OP and MP values are also strongly underestimated as compared to the physical reality in which many sub-resolution pores are in contact with both organic and mineral phases, especially so in clay soil. So again the computed probabilities are estimates based on CT-images, which also only apply to the spatial resolution at hand: The probability of OM to be in contact with a pore >6.6 µm is 0.02-0.03. The actual OP at finer resolution is likely much larger. This needs to be properly explained in the discussion and certainly in Figure 7s caption.*

**Response**: We have included a note in the caption to the Figure which displays the transition probabilities (now Figure 6) that these estimates must be understood at this resolution and that the values might be different at finer scales. We also refer to this in the Results and Discussion sections.

**Comment:** *The value of calculating semi-variogram ranges for all three soil phases is less apparent. I found Fig. 9 overloaded and table 3 redundant. There is really no need to present semi-variogram ranges for the mineral and pore phases so please remove these from Fig. 9 and omit Table 3. You could indicate the median range values (in µm) within Fig. 9.*

**Response**: We have removed the range estimates for mineral and pore phases from Table 3. However, contrary to your comment, we felt that it is useful to present

the ranges for all three phases in (new) Figure 8. We think readers would want to understand how the variations differ among the three phases within aggregates at these fine scales and we comment on this in the Results and discussion section in relation to the vast majority of variation occurring at scales of <250 μm.

**Comment:** *The statement The transition probabilities between OM-centred voxels and adjacent pore voxels a measure of OM accessibility were both small and of limited variation (probabilities between 0.02 and 0.03) for all aggregates and there was no clear relationship between accessibility and the magnitude of SHR. In the conclusions stands a bit isolated. So what is your conclusion now: is it still worthwhile calculating transition probabilities? What about calculating these for transitions in future from OM to specific pore size classes*

**Response**: The transition probability statements have been updated to reflect the new values reported in the revised version.

**Comment:** *The statement shorter length scales of OM variation leads to greater frequencies of microsites and greater CO2 production through microbial respiration and we present preliminary evidence to support this relationship. in the conclusion is interesting. In practice a larger range could represent the presence of more larger particulate OM particles, with many OO transitions over considerable distance (even milimeters). If aggregates contain much particulate OM the OM variation scale / respiration relation could also be inverse: fresh organic substrate particles encapsulated in aggregates (e.g. plant litter) may be more labile, resulting in a higher soil respiration than in a soil aggregate with an equal amount of OM, which is instead mineral-bound. I would be happy if the authors could comment on this in their discussion.*

**Response**: On re-processing the data – using a OM density of 1.4 – we no longer observe the significant relationship between length scale and SHR. We do not therefore consider it appropriate to comment on the potential relationships between length scale, size of organic matter in aggregates and their magnitude of SHR because this would be only speculative.

**Comment:** *R1Com1. I agree to referee 1s main comment. The provided pore shape*

*factor F on its own does not well quantify phenomena like tortuosity. I also follow the authors in that data presented in Figs. 4 and 5 should not be omitted.*

**Response**: We have undertaken new analyses (since the original version) and present data for pore diameter (distribution) and tortuosity in the new version of the paper.

**Comment:** *R1Com4 please add a sentence explaining the purpose of beading the aggregate in glass beads*

**Response**: We have added this sentence to the new version.

**Comment:** *R1Com5 I appreciate the authors initiative to as far as possible enable reproduction of their work by detailing the workflow of the CT volume processing. However, section 2.5.1 was pertinent because of the specific packing of aggregates in quartz beads. The relevance of this section may be small in future experiments with other ways of packing soil samples. So I would ask the authors to move the stepwise listing presented in 2.5.1 to supplementary material.*

**Response**: We have moved the stepwise description to supplementary material file.

**Comment:** *R2Com15 ok with authors response, but could this point not be added to a perspective for a future relevant avenue?* **Response**: We have added a sentence on this in the new version.

[revised manuscript text omitted]